# DISCRETE GRAPH STRUCTURE LEARNING FOR FORECASTING MULTIPLE TIME SERIES

**Chao Shang**[*]
University of Connecticut
chao.shang@uconn.edu

**Jie Chen**[†]
MIT-IBM Watson AI Lab, IBM Research
chenjie@us.ibm.com

**Jinbo Bi**
University of Connecticut
jinbo.bi@uconn.edu

## ABSTRACT

Time series forecasting is an extensively studied subject in statistics, economics, and computer science. Exploration of the correlation and causation among the variables in a multivariate time series shows promise in enhancing the performance of a time series model. When using deep neural networks as forecasting models, we hypothesize that exploiting the pairwise information among multiple (multivariate) time series also improves their forecast. If an explicit graph structure is known, graph neural networks (GNNs) have been demonstrated as powerful tools to exploit the structure. In this work, we propose learning the structure simultaneously with the GNN if the graph is unknown. We cast the problem as learning a probabilistic graph model through optimizing the mean performance over the graph distribution. The distribution is parameterized by a neural network so that discrete graphs can be sampled differentiably through reparameterization. Empirical evaluations show that our method is simpler, more efficient, and better performing than a recently proposed bilevel learning approach for graph structure learning, as well as a broad array of forecasting models, either deep or non-deep learning based, and graph or non-graph based.

## 1 INTRODUCTION

Time series data are widely studied in science and engineering that involve temporal measurements. Time series forecasting is concerned with the prediction of future values based on observed ones in the past. It has played important roles in climate studies, market analysis, traffic control, and energy grid management (Makridakis et al., 1997) and has inspired the development of various predictive models that capture the temporal dynamics of the underlying system. These models range from early autoregressive approaches (Hamilton, 1994; Asteriou & Hall, 2011) to the recent deep learning methods (Seo et al., 2016; Li et al., 2018; Yu et al., 2018; Zhao et al., 2019).

Analysis of univariate time series (a single longitudinal variable) has been extended to multivariate time series and multiple (univariate or multivariate) time series. Multivariate forecasting models find strong predictive power in stressing the interdependency (and even causal relationship) among the variables. The vector autoregressive model (Hamilton, 1994) is an example of multivariate analysis, wherein the coefficient magnitudes offer hints into the Granger causality (Granger, 1969) of one variable to another.

For multiple time series, pairwise similarities or connections among them have also been explored to improve the forecasting accuracy (Yu et al., 2018). An example is the traffic network where each node denotes a time series recording captured by a particular sensor. The spatial connections of the roads offer insights into how traffic dynamics propagates along the network. Several graph neural

---

[*]This work was done while C. Shang was an intern at MIT-IBM Watson AI Lab, IBM Research.
[†]To whom correspondence should be addressed.

network (GNN) approaches (Seo et al., 2016; Li et al., 2018; Yu et al., 2018; Zhao et al., 2019) have been proposed recently to leverage the graph structure for forecasting all time series simultaneously.

The graph structure however is not always available or it may be incomplete. There could be several reasons, including the difficulty in obtaining such information or a deliberate shielding for the protection of sensitive information. For example, a data set comprising sensory readings of the nation-wide energy grid is granted access to specific users without disclosure of the grid structure. Such practical situations incentivize the automatic learning of the hidden graph structure jointly with the forecasting model.

Because GNN approaches show promise in forecasting multiple interrelated time series, in this paper we are concerned with structure learning methods applicable to the downstream use of GNNs. A prominent example is the recent work of Franceschi et al. (2019) (named LDS), which is a meta-learning approach that treats the graph as a hyperparameter in a bilevel optimization framework (Franceschi et al., 2017). Specifically, let $X_{\text{train}}$ and $X_{\text{val}}$ denote the training and the validation sets of time series respectively, $A \in \{0, 1\}^{n \times n}$ denote the graph adjacency matrix of the $n$ time series, $w$ denote the parameters used in the GNN, and $L$ and $F$ denote the the loss functions used during training and validation respectively (which may not be identical). LDS formulates the problem as learning the probability matrix $\theta \in [0, 1]^{n \times n}$, which parameterizes the element-wise Bernoulli distribution from which the adjacency matrix $A$ is sampled:

$$
\begin{aligned}
\min_{\theta} \quad & \mathrm{E}_{A \sim \mathrm{Ber}(\theta)}[F(A, w(\theta), X_{\text{val}})], \\
\text{s.t.} \quad & w(\theta) = \operatorname*{argmin}_{w} \mathrm{E}_{A \sim \mathrm{Ber}(\theta)}[L(A, w, X_{\text{train}})].
\end{aligned}
\tag{1}
$$

Formulation (1) gives a bilevel optimization problem. The constraint (which by itself is an optimization problem) defines the GNN weights as a function of the given graph, so that the objective is to optimize over such a graph only. Note that for differentiability, one does not directly operate on the discrete graph adjacency matrix $A$, but on the continuous probabilities $\theta$ instead.

LDS has two drawbacks. First, its computation is expensive. The derivative of $w$ with respect to $\theta$ is computed by applying the chain rule on a recursive-dynamics surrogate of the inner optimization argmin. Applying the chain rule on this surrogate is equivalent to differentiating an RNN, which is either memory intensive if done in the reverse mode or time consuming if done in the forward mode, when unrolling a deep dynamics. Second, it is challenging to scale. The matrix $\theta$ has $\Theta(n^2)$ entries to optimize and thus the method is hard to scale to increasingly more time series.

In light of the challenges of LDS, we instead advocate a unilevel optimization:

$$
\min_{w} \quad \mathrm{E}_{A \sim \mathrm{Ber}(\theta(w))}[F(A, w, X_{\text{train}})].
\tag{2}
$$

Formulation (2) trains the GNN model as usual, except that the probabilities $\theta$ (which parameterizes the distribution from which $A$ is sampled), is by itself parameterized. We absorb these parameters, together with the GNN parameters, into the notation $w$. We still use a validation set $X_{\text{val}}$ for usual hyperparameter tuning, but these hyperparameters are not $\theta$ as treated by (1). In fact, formulation (1) may need a second validation set to tune other hyperparameters.

The major distinction of our approach from LDS is the parameterization $\theta(w)$, as opposed to an inner optimization $w(\theta)$. In our approach, a modeler owns the freedom to design the parameterization and better control the number of parameters as $n^2$ increases. To this end, time series representation learning and link prediction techniques offer ample inspiration for modeling. In contrast, LDS is more agnostic as no modeling is needed. The effort, instead, lies in the nontrivial treatment of the inner optimization (in particular, its differentiation).

As such, our approach is advantageous in two regards. First, its computation is less expensive, because the gradient computation of a unilevel optimization is straightforward and efficient and implementations are mature. Second, it better scales, because the number of parameters does not grow quadratically with the number of time series.

We coin our approach GTS (short for "graph for time series"), signaling the usefulness of graph structure learning for enhancing time series forecasting. It is important to note that the end purpose of the graph is to improve forecasting quality, rather than identifying causal relationship of the series or recovering the ground-truth graph, if any. While causal discovery of multiple scalar variables is an

established field, identifying causality among multiple multivariate time series requires a nontrivial extension that spans beyond the current study. On the other hand, the graph, either learned or pre-existing, serves as additional information that helps the model better capture global signals and apply on each series. There does not exist a golden measure for the quality of the learned graph except forecasting accuracy. For example, the traffic network does not necessarily offer the best pairwise relationship a GNN can exploit for forecasting traffic series. Nevertheless, to robustify GTS we incorporate regularization that penalizes significant departure from one's prior belief. If a certain "ground-truth" graph is believed, the learned graph will be a healthy variation of it for a more accurate forecast.

## 2 RELATED WORK

Time series forecasting has been studied for decades by statisticians. It is out of the scope of this paper to comprehensively survey the literature, but we will focus more on late developments under the deep learning context. Early textbook methods include (vector) autoregressive models (Hamilton, 1994), autoregressive integrated moving average (ARIMA) (Asteriou & Hall, 2011), hidden Markov models (HMM) (Baum & Petrie, 1966), and Kalman filters (Zarchan & Musoff, 2000). Generally speaking, these are linear models that use a window of the past information to predict the next time step, although nonlinear versions with parameterization are subsequently developed.

A notable nonlinear extension was the RNN (Williams et al., 1986), which later evolved into LSTM (Hochreiter & Schmidhuber, 1997), BiLSTM (Schuster & Paliwal, 1997), and GRU (Cho et al., 2014), which addressed several limitations of the vanilla RNN, such as the vanishing gradient problem. These architectures are hard to parallelize because of the recurrent nature of the forward and backward computation. More recently, Transformer (Vaswani et al., 2017) and BERT (Devlin et al., 2019) were developed to address parallelization, by introducing attention mechanisms that simultaneously digested past (and future) information. Although these models are more heavily used for sequence data under the context of natural language processing, they are readily applicable for time series as well (Shih et al., 2019; Li et al., 2019).

Graph neural networks (Zhang et al., 2018; Zhou et al., 2018; Wu et al., 2019) emerged quickly in deep learning to handle graph-structured data. Typically, graph nodes are represented by feature vectors, but for the case of time series, a number of specialized architectures were recently developed; see, e.g., GCRN (Seo et al., 2016), DCRNN (Li et al., 2018), STGCN (Yu et al., 2018), and T-GCN (Zhao et al., 2019). These architectures essentially combine the temporal recurrent processing with graph convolution to augment the representation learning of the individual time series.

Graph structure learning (not necessarily for time series) appears in various contexts and thus methods span a broad spectrum. One field of study is probabilistic graphical models and casual inference, whereby the directed acyclic structure is enforced. Gradient-based approaches in this context include NOTEARS (Zheng et al., 2018), DAG-GNN (Yu et al., 2019), and GraN-DAG (Lachapelle et al., 2020). On the other hand, a general graph may still be useful without resorting to causality. LDS (Franceschi et al., 2019) is a meta-learning approach that demonstrates to improve the performance on node classification tasks. MTGNN (Wu et al., 2020) parameterizes the graph as a degree-$k$ graph, which is learned end-to-end with a GNN for forecasting time series. We, on the other hand, allow a more general structural prior for the graph. NRI (Kipf et al., 2018) adopts a latent-variable approach and learns a latent graph for forecasting system dynamics. Our approach is closely related to NRI and we will compare with it in the following section after introducing the technical details.

## 3 METHOD

In this section, we present the proposed GTS method, elaborate the model parameterization, and describe the training technique. We also highlight the distinctions from NRI (Kipf et al., 2018).

Let us first settle the notations. Denote by $X$ the training data, which is a three dimensional tensor, with the three dimensions being feature, time, and the $n$ series. Superscript refers to the series and subscript refers to time; that is, $X^i$ denotes the $i$-th series for all features and time and $X_t$ denotes the $t$-th time step for all features and series. There are in total $S$ time steps for training. The model will use a window of $T$ steps to forecast the next $\tau$ steps. For each valid $t$, denote by

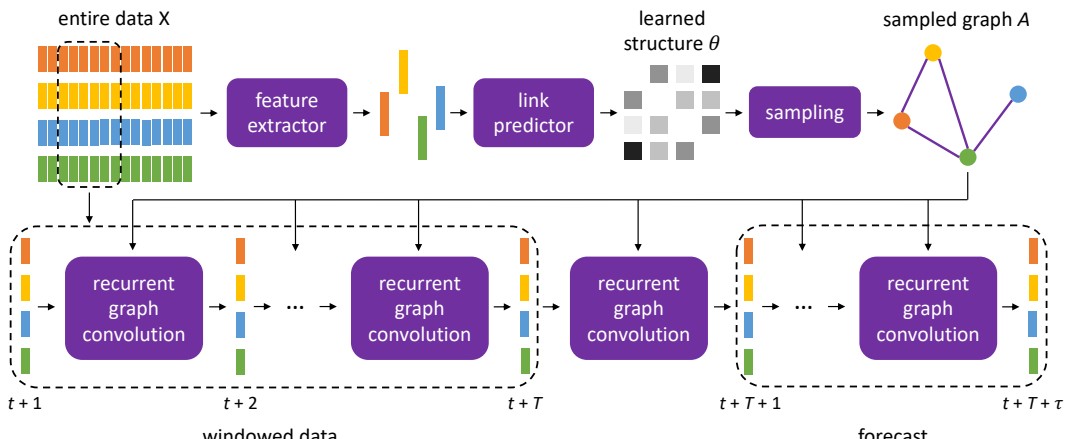

Figure 1: GTS architecture.

$\widehat{X}_{t+T+1:t+T+\tau} = f(A, w, X_{t+1:t+T})$ the model, which forecasts $\widehat{X}_{t+T+1:t+T+\tau}$ from observations $X_{t+1:t+T}$, through exploiting the graph structure $A$ and being parameterized by $w$. Using $\ell$ to denote the loss function between the prediction and the ground truth, a typical training objective reads

$$\sum_t \ell(f(A, w, X_{t+1:t+T}), \ X_{t+T+1:t+T+\tau}). \tag{3}$$

Three remaining details are the parameterization of $A$, the model $f$, and the loss $\ell$.

### 3.1 GRAPH STRUCTURE PARAMETERIZATION

The binary matrix $A \in \{0, 1\}^{n \times n}$ by itself is challenging to parameterize, because it requires a differentiable function that outputs discrete values 0/1. A natural idea is to let $A$ be a random variable of the matrix Bernoulli distribution parameterized by $\theta \in [0, 1]^{n \times n}$, so that $A_{ij}$ is independent for all the $(i, j)$ pairs with $A_{ij} \sim \text{Ber}(\theta_{ij})$. Here, $\theta_{ij}$ is the success probability of a Bernoulli distribution. Then, the training objective (3) needs to be modified to

$$\mathrm{E}_{A \sim \text{Ber}(\theta)} \left[ \sum_t \ell(f(A, w, X_{t+1:t+T}), \ X_{t+T+1:t+T+\tau}) \right]. \tag{4}$$

As hinted in Section 1, we further parameterize $\theta$ as $\theta(w)$, because otherwise the $n^2$ degrees of freedom in $\theta$ render the optimization hard to scale. Such a parameterization, however, imposes a challenge on differentiability, if the expectation (4) is evaluated through sample average: the gradient of (4) does not flow through $A$ in a usual Bernoulli sampling. Hence, we apply the Gumbel reparameterization trick proposed by Jang et al. (2017) and Maddison et al. (2017): $A_{ij} = \text{sigmoid}((\log(\theta_{ij}/(1 - \theta_{ij})) + (g_{ij}^1 - g_{ij}^2))/s)$, where $g_{ij}^1, g_{ij}^2 \sim \text{Gumbel}(0, 1)$ for all $i, j$. When the temperature $s \to 0$, $A_{ij} = 1$ with probability $\theta_{ij}$ and 0 with remaining probability. In practice, we anneal $s$ progressively in training such that it tends to zero.

For the parameterization of $\theta$, we use a feature extractor to yield a feature vector for each series and a link predictor that takes in a pair of feature vectors and outputs a link probability. The feature extractor maps a matrix $X^i$ to a vector $z^i$ for each $i$. Many sequence architectures can be applied; we opt for a simple one. Specifically, we perform convolution along the temporal dimension, vectorize along this dimension, and apply a fully connected layer to reduce the dimension; that is, $z^i = \text{FC}(\text{vec}(\text{Conv}(X^i)))$. Note that the feature extractor is conducted on the entire sequence rather than a window of $T$ time steps. Weights are shared among all series.

The link predictor maps a pair of vectors $(z^i, z^j)$ to a scalar $\theta_{ij} \in [0, 1]$. We concatenate the two vectors and apply two fully connected layers to achieve so; that is, $\theta_{ij} = \text{FC}(\text{FC}(z^i \| z^j))$. The last activation needs be a sigmoid. See the top part of Figure 1.

### 3.2 GRAPH NEURAL NETWORK FORECASTING

The bottom part of Figure 1 is the forecasting model $f$. We use a sequence-to-sequence (seq2seq) model (Sutskever et al., 2014) to map $X_{t+1:t+T}^i$ to $X_{t+T+1:t+T+\tau}^i$ for each series $i$. Seq2seq is

typically a recurrent model, but with a graph structure available among the series, we leverage recurrent graph convolution to handle all series simultaneously, as opposed to the usual recurrent mechanism that treats each series separately.

Specifically, for each time step $t'$, the seq2seq model takes $X_{t'}$ for all series as input and updates the internal hidden state from $H_{t'-1}$ to $H_{t'}$. The encoder part of the seq2seq performs recurrent updates from $t' = t + 1$ to $t' = t + T$, producing $H_{t+T}$ as a summary of the input. The decoder part uses $H_{t+T}$ to continue the recurrence and evolves the hidden state for another $\tau$ steps. Each hidden state $H_{t'}, t' = t + T + 1 : t + T + \tau$, simultaneously serves as the output $\widehat{X}_{t'}$ and the input to the next time step.

The recurrence that accepts input and updates hidden states collectively for all series uses a graph convolution to replace the usual multiplication with a weight matrix. Several existing architectures serve this purpose (e.g., GCRN (Seo et al., 2016), STGCN (Yu et al., 2018), and T-GCN (Zhao et al., 2019)), but we use the diffusion convolutional GRU defined in DCRNN (Li et al., 2018) because it is designed for directed graphs:

$$R_{t'} = \text{sigmoid}(W_R \star_A [X_{t'} \,\|\, H_{t'-1}] + b_R), \quad C_{t'} = \tanh(W_C \star_A [X_{t'} \,\|\, (R_{t'} \odot H_{t'-1}] + b_C),$$
$$U_{t'} = \text{sigmoid}(W_U \star_A [X_{t'} \,\|\, H_{t'-1}] + b_U), \quad H_{t'} = U_{t'} \odot H_{t'-1} + (1 - U_{t'}) \odot C_{t'},$$

where the graph convolution $\star_A$ is defined as

$$W_Q \star_A Y = \sum_{k=0}^{K} \left( w_{k,1}^Q (D_O^{-1} A)^k + w_{k,2}^Q (D_I^{-1} A^T)^k \right) Y,$$

with $D_O$ and $D_I$ being the out-degree and in-degree matrix and $\|$ being concatenation along the feature dimension. Here, $w_{k,1}^Q$, $w_{k,2}^Q$, $b_Q$ for $Q = R, U, C$ are model parameters and the diffusion degree $K$ is a hyperparameter.

We remark that as a subsequent experiment corroborates, this GNN model can be replaced by other similar ones (e.g., T-GCN), such that the forecast performance remains similar while still being superior over all baselines. In comparison, the more crucial part of our proposal is the structure learning component (presented in the preceding subsection), without which it falls back to a model either using no graphs or needing a supplied one, both performing less well.

## 3.3 Training, Optionally with a Priori Knowledge of the Graph

The base training loss (per window) is the mean absolute error between the forecast and the ground truth

$$\ell_{\text{base}}^t(\widehat{X}_{t+T+1:t+T+\tau}, \; X_{t+T+1:t+T+\tau}) = \tfrac{1}{\tau} \sum_{t'=t+T+1}^{t+T+\tau} |\widehat{X}_{t'} - X_{t'}|.$$

Additionally, we propose a regularization that improves graph quality, through injecting a priori knowledge of the pairwise interaction into the model. Sometimes an actual graph among the time series is known, such as the case of traffic network mentioned in Section 1. Generally, even if an explicit structure is unknown, a neighborhood graph (such as a $k$NN graph) may still serve as reasonable knowledge. The use of $k$NN encourages sparsity if $k$ is small, which circumvents the drawback of $\ell_1$ constraints that cannot be easily imposed because the graph is not a raw variable to optimize. As such, we use the cross-entropy between $\theta$ and the a priori graph $A^a$ as the regularization:

$$\ell_{\text{reg}} = \sum_{ij} -A_{ij}^a \log \theta_{ij} - (1 - A_{ij}^a) \log(1 - \theta_{ij}). \tag{5}$$

The overall training loss is then $\sum_t \ell_{\text{base}}^t + \lambda \ell_{\text{reg}}$, with $\lambda > 0$ being the regularization magnitude.

## 3.4 Comparison with NRI

GTS appears similar to NRI (Kipf et al., 2018) on the surface, because both compute a pairwise structure from multiple time series and use the structure to improve forecasting. In these two methods, the architecture to compute the structure, as well as the one to forecast, bare many differences; but these differences are only secondary. The most essential distinction is the number of structures. To avoid confusion, here we say "structure" ($\theta$) rather than "graph" ($A$) because there are combinatorially many graph samples from the same structure. Our approach produces one single structure given one set of $n$ series. On the contrary, the autoencoder approach adopted by NRI produces different structures given different encoding inputs. Hence, a feasible use of NRI can only occur in the

following two manners. (a) A single set of $n$ series is given and training is done on windowed data, where each window will produce a separate structure. (b) Many sets are given and training is done through iterating each set, which corresponds to a separate structure. Both cases are different from our scenario, where a single set of time series is given and a single structure is produced.

Fundamentally, NRI is a variational autoencoder and thus the inference of the structure is an amortized inference: under setting (b) above, the inferred structure is a posterior given a set of series. The amortization uses an encoder parameterization to free off the tedious posterior inference whenever a new set of series arrives. Moreover, under the evidence lower bound (ELBO) training objective, the prior is a graph, each edge of which takes a value uniformly in $[0, 1]$. In our case, on the contrary, a single structure is desired. Thus, amortized inference is neither necessary nor relevant. Furthermore, one may interpret the a priori information $A^{\mathrm{a}}$ for regularization as a "structural prior;" however, for each node pair it offers a stronger preference on the existence/absence of an edge than a uniform probability.

## 4 EXPERIMENTS

In this section, we conduct extensive experiments to show that the proposed method GTS outperforms a comprehensive set of forecasting methods, including one that learns a hidden graph structure (LDS, adapted for time series). We also demonstrate that GTS is computationally efficient and is able to learn a graph close to the a priori knowledge through regularization, with little compromise on the forecasting quality.

### 4.1 SETUP

**Data sets.** We experiment with two benchmark data sets METR-LA and PEMS-BAY from Li et al. (2018) and a proprietary data set PMU. The first two are traffic data sets with given graphs serving as ground truths; we perform no processing and follow the same configuration as in the referenced work for experimentation. The last one is a sensor network of the U.S. power grid without a given grid topology. For details, see Appendix Section A. For all data sets, we use a temporal 70/10/20 split for training, validation, and testing, respectively.

**Baselines.** We compare with a number of forecasting methods:

1. Non-deep learning methods: historical average (HA), ARIMA with Kalman filter (ARIMA), vector auto-regression (VAR), and support vector regression (SVR). The historical average accounts for weekly seasonality and predicts for a day by using the weighted average of the same day in the past few weeks.

2. Deep learning methods that treat each series separately (i.e., no graph): feed-forward neural network (FNN) and LSTM.

3. GNN method applied on the given graph (or $k$NN graph for PMU): DCRNN (Li et al., 2018).

4. GNN methods that simultaneously learn a graph structure. We use LDS (Franceschi et al., 2019) to learn the graph, wherein the forecast model is DCRNN. We name the method "LDS" for short. Additionally, we compare with NRI (Kipf et al., 2018).

5. Variant of GTS: We maintain the graph structure parameterization but replace the DCRNN forecast model by T-GCN (Zhao et al., 2019). We name the variant "GTSv."

Except LDS and NRI, all baselines follow the configurations presented in Li et al. (2018). For LDS, we follow Franceschi et al. (2019). For NRI, we follow Kipf et al. (2018).

**Evaluation metrics.** All methods are evaluated with three metrics: mean absolute error (MAE), root mean square error (RMSE), and mean absolute percentage error (MAPE).

For details on hyperparameter setting and training platform, see Appendix Section B.

## 4.2 RESULTS

**Forecasting quality.** We first evaluate the performance of GTS through comparing it with all the aforementioned baselines. Because of the significant memory consumption of NRI, this method is executed on only the smaller data set PMU. The tasks are to forecast 15, 30, and 60 minutes.

Table 1: Forecasting error (METR-LA).

|        | Metric | HA | ARIMA | VAR | SVR | FNN | LSTM | DCRNN | LDS | GTSv | GTS |
|--------|--------|------|-------|-------|-------|-------|------|-------|-------|------|------|
| 15 min | MAE | 4.16 | 3.99 | 4.42 | 3.99 | 3.99 | 3.44 | 2.77 | 2.75 | 2.61 | **2.39** |
|        | RMSE | 7.80 | 8.21 | 7.89 | 8.45 | 7.94 | 6.30 | 5.38 | 5.35 | 4.83 | **4.41** |
|        | MAPE | 13.0% | 9.6% | 10.2% | 9.3% | 9.9% | 9.6% | 7.3% | 7.1% | 6.8% | **6.0%** |
| 30 min | MAE | 4.16 | 5.15 | 5.41 | 5.05 | 4.23 | 3.77 | 3.15 | 3.14 | 2.94 | **2.65** |
|        | RMSE | 7.80 | 10.45 | 9.13 | 10.87 | 8.17 | 7.23 | 6.45 | 6.45 | 5.62 | **5.06** |
|        | MAPE | 13.0% | 12.7% | 12.7% | 12.1% | 12.9% | 10.9% | 8.8% | 8.6% | 7.9% | **7.0%** |
| 60 min | MAE | 4.16 | 6.90 | 6.52 | 6.72 | 4.49 | 4.37 | 3.60 | 3.63 | 3.46 | **2.99** |
|        | RMSE | 7.80 | 13.23 | 10.11 | 13.76 | 8.69 | 8.69 | 7.59 | 7.67 | 6.67 | **5.85** |
|        | MAPE | 13.0% | 17.4% | 15.8% | 16.7% | 14.0% | 13.2% | 10.5% | 10.34% | 9.9% | **8.3%** |

Table 1 summarizes the results for METR-LA. A few observations follow. (1) Deep learning methods generally outperform non-deep learning methods, except historical average that performs on par with deep learning in some metrics. Seasonality is a strong indicator of the repeating traffic patterns and not surprisingly HA performs reasonably well despite simplicity. (2) Among the deep learning methods, graph-based models outperform non-graph models. This result corroborates the premise of this work: graph structure is helpful. (3) Among the graph-based methods, LDS performs slightly better than DCRNN. The difference between these two methods is that the latter employs the given graph, which may or may not imply direct interactions, whereas the former learns a graph in the data-driven manner. Their performances however are quite similar. (4) The most encouraging result is that the proposed method GTS significantly outperforms LDS and hence DCRNN. GTS learns a graph structure through parameterization, rather than treating it as a (hyper)parameter which is the case in LDS. (5) The performance of the variant GTSv stays between GTS and LDS. This observation corroborates that the proposed structure learning component contributes more crucially to the overall performance than does a careful choice of the GNN forecasting component.

To dive into the behavioral difference between GTS and DCRNN, we plot in Figure 2 two forecasting examples. One sees that both methods produce smooth series. In the top example, overall the GTS curve is closer to the moving average of the ground truth than is the DCRNN curve (see e.g., the left part and the U shape). In the bottom example, the GTS curve better captures the sharp dip toward the end of the series. In both examples, there exist several short but deep downward spikes. Such anomalous data are captured by neither methods.

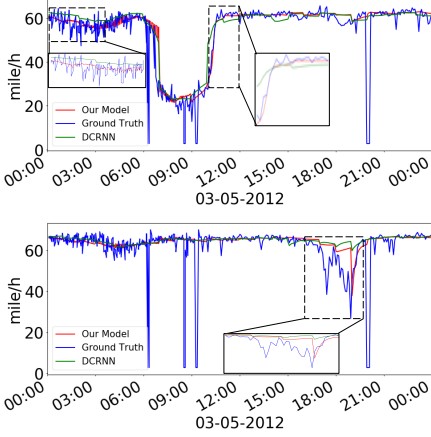

Figure 2: One-day forecast (METR-LA).

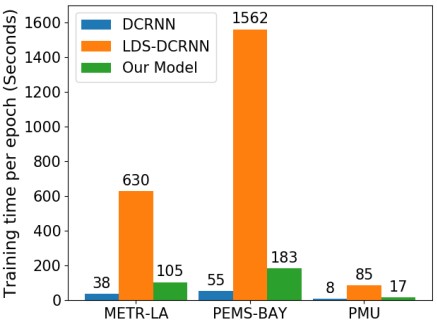

Figure 3: Training time per epoch. Not learning a graph (DCRNN) is the fastest to train and learning a graph by using LDS needs orders of magnitude more time. Our model learns a graph with a favorable overhead.

Additionally, we summarize the results for PEMS-BAY and PMU in Table 3 and 2, respectively. (see Appendix Section C for the former). The observations are rather similar to those of METR-LA. Our model produces the best prediction in all scenarios and under all metrics. Additionally, for the PMU data set, NRI performs competitively, second to GTS/GTSv and better than LDS in most of the cases.

Table 2: Forecasting error (PMU).

|  | Metric | FNN | LSTM | DCRNN | LDS | NRI | GTSv | GTS |
|---|---|---|---|---|---|---|---|---|
| **15 min** | MAE ($\times 10^{-3}$) | 1.23 | 1.02 | 0.71 | 0.49 | 0.66 | 0.26 | **0.24** |
|  | RMSE ($\times 10^{-2}$) | 1.28 | 1.63 | 1.42 | 1.26 | 0.27 | 0.20 | **0.19** |
|  | MAPE | 0.20% | 0.21% | 0.09% | 0.07% | 0.14% | 0.05% | **0.04%** |
| **30 min** | MAE ($\times 10^{-3}$) | 1.42 | 1.11 | 1.08 | 0.81 | 0.71 | 0.31 | **0.30** |
|  | RMSE ($\times 10^{-2}$) | 1.81 | 2.06 | 1.91 | 1.79 | 0.30 | 0.23 | **0.22** |
|  | MAPE | 0.23% | 0.20% | 0.15% | 0.12% | 0.15% | **0.05%** | **0.05%** |
| **60 min** | MAE ($\times 10^{-3}$) | 1.88 | 1.79 | 1.78 | 1.45 | 0.83 | **0.39** | 0.41 |
|  | RMSE ($\times 10^{-2}$) | 2.58 | 2.75 | 2.65 | 2.54 | 0.46 | 0.32 | **0.30** |
|  | MAPE | 0.29% | 0.27% | 0.24% | 0.22% | 0.17% | **0.07%** | **0.07%** |

**Computational efficiency.** We compare the training costs of the graph-based methods: DCRNN, LDS, and GTS. See Figure 3. DCRNN is the most efficient to train, since no graph structure learning is involved. To learn the graph, LDS needs orders of magnitude more time than does DCRNN. Recall that LDS employs a bilevel optimization (1), which is computationally highly challenging. In contrast, the proposed method GTS learns the graph structure as a byproduct of the model training (2). Its training time is approximately three times of that of DCRNN, a favorable overhead compared with the forbidding cost of LDS.

**Effect of regularization.** We propose in Section 3.3 using regularization to incorporate a priori knowledge of the graph. One salient example of knowledge is sparsity, which postulates that many node pairs barely interact. We show the effect of regularization on the data set PMU with the use of a $k$NN graph as knowledge. The task is 15-minute forecasting and results (expected degree and MAE) are plotted in Figure 4. The decreasing curves in both plots indicate that using a smaller $k$ or increasing the regularization magnitude produces sparser graphs. The bars give the MAEs, all around 2.4e-4, indicating equally good forecasting quality. (Note that the MAE for LDS is 4.9e-4.)

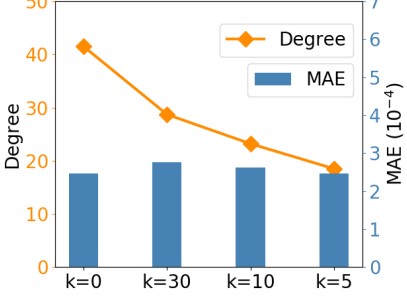
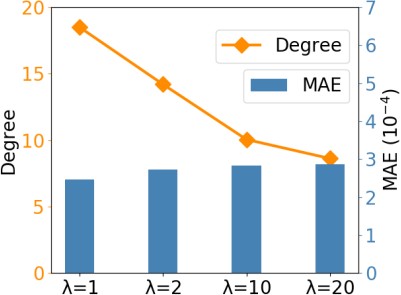

(a) Fix regularization magnitude $= 1$ and vary $k$. The case $k = 0$ means no $k$NN regularization.

(b) Fix $k = 5$ and vary regularization $\lambda$.

Figure 4: Effect of regularization (PMU). "Degree" means expected degree of the graph.

**Learned structures.** To examine the learned structure $\theta$, we further show its difference from the given graph adjacency matrix $A^{\mathrm{a}}$ (binary) and visualize one particular example in Figure 5. The difference is defined as $\ell_{\mathrm{reg}}/n^2$ (average cross entropy; see (5)). One reads that when $\lambda = 20$, the difference is 0.34. It indicates that the learned probabilities in $\theta$ are on average 0.3 away from the entries of $A^{\mathrm{a}}$, because $-\log(1 - 0.3) \approx 0.34$. When using 0.5 as a cutoff threshold for $\theta$, such a difference possibly results in false-positive edges (existing in $\theta$ but not in $A^{\mathrm{a}}$; orange dashed) and false-negative edges (existing in $A^{\mathrm{a}}$ but not in $\theta$; none in the example).

Note that the regularization strength $\lambda$ weighs the forecasting error (MAE) and the cross entropy in the loss function. When $\lambda = 0$, the training loss is not regularized, yielding optimal forecast results reported in Table 2. When $\lambda = \infty$, one effectively enforces $\theta$ to be identical to $A^{\mathrm{a}}$ and hence the model reduces to DCRNN, whose forecasting performance is worse than our model. The interesting question is when $\lambda$ interpolates between the two extremes, whether it is possible to find a sweet spot such that forecasting performance is close to our model but meanwhile $\theta$ is close to $A^{\mathrm{a}}$. Figure 5 suggests positively. We stress that our model does not intend to learn a "ground-truth" graph (e.g., the traffic network or the power grid); but rather, learn a structure that a GNN can exploit to improve forecast.

|        |     | $\lambda = 1$ | $\lambda = 2$ | $\lambda = 10$ | $\lambda = 20$ |
|--------|-----|---------------|---------------|----------------|----------------|
| 15 min | CE  | 1.93          | 1.87          | 0.53           | 0.34           |
|        | MAE | 2.47e-4       | 2.74e-4       | 2.83e-4        | 2.87e-4        |
| 30 min | CE  | 1.93          | 1.87          | 0.53           | 0.34           |
|        | MAE | 3.02e-4       | 3.26e-4       | 3.44e-4        | 3.59e-4        |
| 60 min | CE  | 1.93          | 1.87          | 0.53           | 0.34           |
|        | MAE | 4.14e-4       | 4.33e-4       | 4.78e-4        | 5.12e-4        |

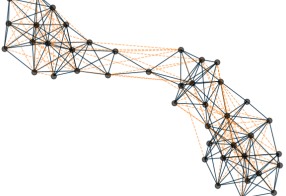

Figure 5: Learned structures (PMU). Left: difference between $A^{\mathrm{a}}$ and $\theta$ (in terms of average cross entropy) as $\lambda$ varies. Right: $A^{\mathrm{a}}$ overlaid with $\theta$ (orange edges exist in $\theta$ but not in $A^{\mathrm{a}}$) when $\lambda = 20$.

**Other structural priors.** In the PMU data set, we use a synthetic $k$NN structure prior $A^{\mathrm{a}}$ due to the lack of a known graph. For METR-LA and PEMS-BAY, however, such as graph can be constructed based on spatial proximity (Li et al., 2018). We show in Figure 6 the effect of regularization for these data sets. Similar to the findings of PMU, moving $\lambda$ between 0 and $\infty$ interpolates two extremes: the best forecast quality and recovery of $A^{\mathrm{a}}$. With a reasonable choice of $\lambda$ (e.g., 0.3), the forecast quality degrades only slightly but the learned structure is rather close to the given $A^{\mathrm{a}}$, judged from the average cross entropy.

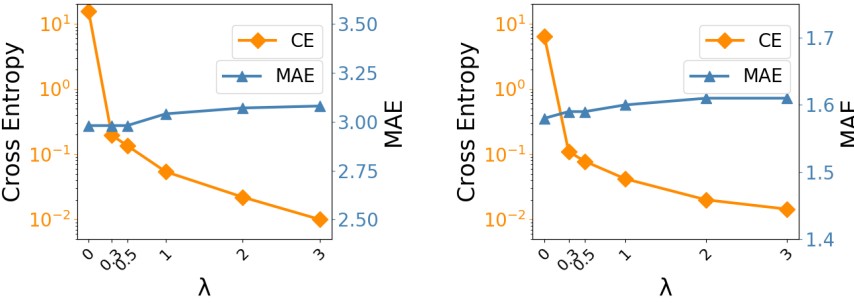

Figure 6: Effect of regularization (left: METR-LA; right: PEMS-BAY; 60 min forecast). Average cross entropy measures departure from $A^{\mathrm{a}}$.

## 5 CONCLUSIONS

We have presented a time series forecasting model that learns a graph structure among multiple time series and forecasts them simultaneously with a GNN. Both the graph and the GNN are learned end-to-end, maximally exploiting the pairwise interactions among data streams. The graph structure is parameterized by neural networks rather than being treated as a (hyper)parameter, hence significantly reducing the training cost compared with a recently proposed bilevel optimization approach LDS. We conduct comprehensive comparisons with a number of baselines, including non-deep learning methods and deep learning methods (which either ignore the pairwise interaction, use a given graph, or learn a graph by using LDS), and show that our approach attains the best forecasting quality. We also demonstrate that regularization helps incorporate a priori knowledge, rendering the learned graph a healthy variation of the given one for more accurate forecast.

ACKNOWLEDGMENT AND DISCLAIMER

This material is based upon work supported by the Department of Energy under Award Number(s) DE-OE0000910. C. Shang was also supported by National Science Foundation grant IIS-1718738 (to J. Bi) during this work. J. Bi was additionally supported by National Institutes of Health grants K02-DA043063 and R01-DA051922. This report was prepared as an account of work sponsored by agencies of the United States Government. Neither the United States Government nor any agency thereof, nor any of their employees, makes any warranty, express or implied, or assumes any legal liability or responsibility for the accuracy, completeness, or usefulness of any information, apparatus, product, or process disclosed, or represents that its use would not infringe privately owned rights. Reference herein to any specific commercial product, process, or service by trade name, trademark, manufacturer, or otherwise does not necessarily constitute or imply its endorsement, recommendation, or favoring by the United States Government or any agency thereof. The views and opinions of authors expressed herein do not necessarily state or reflect those of the United States Government or any agency thereof.

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

# Appendices

## A   ADDITIONAL DETAILS OF DATA SETS

**METR-LA** is a traffic data set collected from loop detectors in the highway of Los Angles, CA (Jagadish et al., 2014). It contains 207 sensors, each of which records four months of data at the frequency of five minutes. A graph of sensors is given; it was constructed by imposing a radial basis function on the pairwise distance of sensors at a certain cutoff. For more information see Li et al. (2018). We perform no processing and follow the same configuration as in Li et al. (2018)

**PEMS-BAY** is also a traffic data set, collected by the California Transportation Agencies Performance Measurement System. It includes 325 sensors in the Bay Area for a period of six months, at the same five-minute frequency. Construction of the graph is the same as that of METR-LA. No processing is performed.

**PMU** contains time series data recorded by the phasor measurement units (PMUs) deployed across the U.S. power grid. We extract one month of data (February 2017) from one interconnect of the grid, which includes 42 PMU sources. Each PMU records a number of state variables and we use only the voltage magnitude and the current magnitude. The PMUs sample the system states at high rates (either 30 or 60 Hertz). We aggregate every five minutes, yielding a data frequency the same as the above two data sets. Different from them, this data set offers neither the grid topology, the sensor identities, nor the sensor locations. Hence, a "ground truth" graph is unknown.

However, it is highly plausible that the PMUs interact in a nontrivial manner, since some series are highly correlated whereas others not much. Figure 7 shows three example series. Visually, the first series appears more correlated to the second one than to the third one. For example, in the first two series, the blue curves (the variable ip_m) are visually seasonal and synchronous. Moreover, inside the purple window, the red curves (the variable vp_m) in the first two series show three downward spikes, which are missing in the third series. Indeed, the correlation matrix between the first two series is $\left( \begin{smallmatrix} 0.76 & -0.04 \\ -0.31 & 0.96 \end{smallmatrix} \right)$ and that between the first and the third series is $\left( \begin{smallmatrix} 0.18 & -0.10 \\ 0.22 & 0.22 \end{smallmatrix} \right)$. Such an observation justifies graph structure learning among the PMUs.

It is important to note a few processing steps of the data set because of its noisy and incomplete nature. The data set contains a fair amount of unreliable readings (e.g., outliers). Hence, we consult domain experts and set lower and upper bounds to filter out extremely small and large values. Accounting for missing data, within every five minutes we take the mean of the available readings if any, or impute with the mean of the entire series.

## B   ADDITIONAL DETAILS OF EXPERIMENT SETTING

**Hyperparameters.** Several hyperparameters are tuned through grid search: initial learning rate $\{0.1, 0.01, 0.001\}$, dropout rate $\{0.1, 0.2, 0.3\}$, embedding size of LSTM $\{32, 64, 128, 256\}$, the $k$ value in $k$NN $\{5, 10, 20, 30\}$, and the weight of regularization $\{0, 1, 2, 5, 10, 20\}$. For other hyperparameters, the convolution kernel size in the feature extractor is 10 and the decay ratio of learning rate is 0.1. After tuning, the best initial learning rate for METR-LA and PEMS-BAY is 0.01 and for PMU is 0.001. The optimizer is Adam.

Because the loss function is an expectation (see (1) and (2)), the expectation is computed as an average of 10 random samples. Such an averaging is needed only for model evaluation. In training, one random sample suffices because the optimizer is a stochastic optimizer.

**Platform.** We implement the models in PyTorch. All experiments are run on one compute node of an IBM Power9 server. The compute node contains 80 CPU cores and four NVidia V100 GPUs, but because of scheduling limitation we use only one GPU.

Code is available at `https://github.com/chaoshangcs/GTS`.

## C   Additional Results for Forecasting Quality

See Table 3 for the forecast results of PEMS-BAY. The observations are rather similar to those of Tables 1 and 2 in Section 4.2. In particular, GTS produces the best prediction in all scenarios and under all metrics.

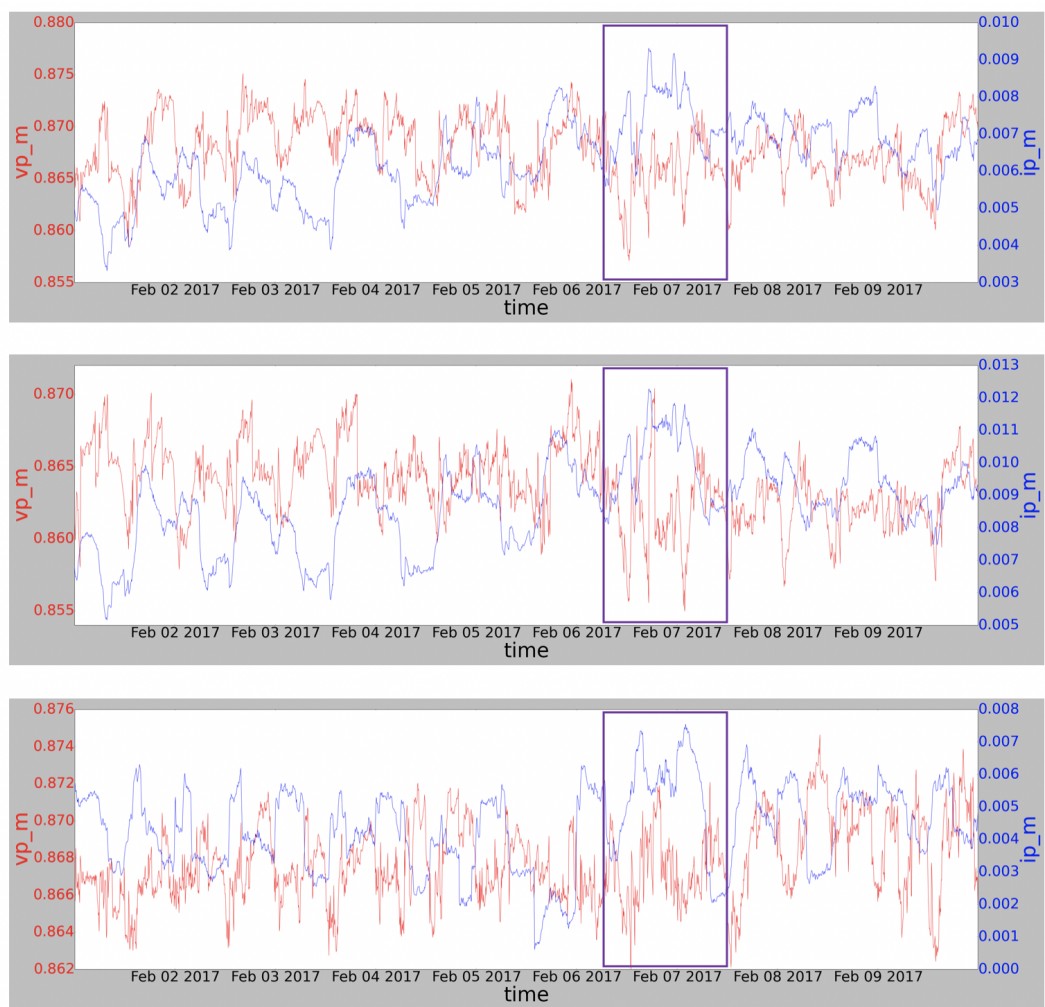

Figure 7: Example time series from the PMU data set. Data have been standardized and the vertical axes do not show the raw value.

Table 3: Forecasting error (PEMS-BAY).

|  | Metric | HA | ARIMA | VAR | SVR | FNN | LSTM | DCRNN | LDS | GTSv | GTS |
|---|---|---|---|---|---|---|---|---|---|---|---|
| 15 min | MAE | 2.88 | 1.62 | 1.74 | 1.85 | 2.20 | 2.05 | 1.38 | 1.33 | 1.16 | **1.12** |
|  | RMSE | 5.59 | 3.30 | 3.16 | 3.59 | 4.42 | 4.19 | 2.95 | 2.81 | 2.27 | **2.16** |
|  | MAPE | 6.8% | 3.5% | 3.6% | 3.8% | 5.2% | 4.8% | 2.9% | 2.8% | **2.3%** | **2.3%** |
| 30 min | MAE | 2.88 | 2.33 | 2.32 | 2.48 | 2.30 | 2.20 | 1.74 | 1.67 | 1.44 | **1.34** |
|  | RMSE | 5.59 | 4.76 | 4.25 | 5.18 | 4.63 | 4.55 | 3.97 | 3.80 | 2.97 | **2.74** |
|  | MAPE | 6.8% | 5.4% | 5.0% | 5.5% | 5.4% | 5.2% | 3.9% | 3.8% | 3.1% | **2.9%** |
| 60 min | MAE | 2.88 | 3.38 | 2.93 | 3.28 | 2.46 | 2.37 | 2.07 | 1.99 | 1.81 | **1.58** |
|  | RMSE | 5.59 | 6.50 | 5.44 | 7.08 | 4.98 | 4.96 | 4.74 | 4.59 | 3.78 | **3.30** |
|  | MAPE | 6.8% | 8.3% | 6.5% | 8.0% | 5.9% | 5.7% | 4.9% | 4.8% | 4.1% | **3.6%** |

# D UPDATES OF TABLES 1, 2, AND 3

Our implementation had been developed based on the PyTorch version of DCRNN (`https://github.com/chnsh/DCRNN_PyTorch`). It was brought to our attention recently that this version calculated the evaluation metrics MAE/RMSE/MAPE in a manner slightly different from that used to report the results of DCRNN in the official publication (`https://github.com/chnsh/DCRNN_PyTorch/issues/3`). We updated Tables 1, 2, and 3 by correcting the calculations to be consistent with the official DCRNN results. See Tables 4, 5, and 6. Despite the correction, observations and conclusions regarding the comparison of different methods remain unchanged.

Table 4: (Correction of Table 1) Forecasting error (METR-LA).

|  | Metric | HA | ARIMA | VAR | SVR | FNN | LSTM | DCRNN | LDS | GTSv | GTS |
|---|---|---|---|---|---|---|---|---|---|---|---|
| 15 min | MAE | 4.16 | 3.99 | 4.42 | 3.99 | 3.99 | 3.44 | 2.77 | 2.75 | 2.74 | **2.64** |
|  | RMSE | 7.80 | 8.21 | 7.89 | 8.45 | 7.94 | 6.30 | 5.38 | 5.35 | 5.09 | **4.95** |
|  | MAPE | 13.0% | 9.6% | 10.2% | 9.3% | 9.9% | 9.6% | 7.3% | 7.1% | 7.3% | **6.8%** |
| 30 min | MAE | 4.16 | 5.15 | 5.41 | 5.05 | 4.23 | 3.77 | 3.15 | 3.14 | 3.11 | **3.01** |
|  | RMSE | 7.80 | 10.45 | 9.13 | 10.87 | 8.17 | 7.23 | 6.45 | 6.45 | 6.02 | **5.85** |
|  | MAPE | 13.0% | 12.7% | 12.7% | 12.1% | 12.9% | 10.9% | 8.8% | 8.6% | 8.7% | **8.2%** |
| 60 min | MAE | 4.16 | 6.90 | 6.52 | 6.72 | 4.49 | 4.37 | 3.60 | 3.63 | 3.53 | **3.41** |
|  | RMSE | 7.80 | 13.23 | 10.11 | 13.76 | 8.69 | 8.69 | 7.59 | 7.67 | 6.84 | **6.74** |
|  | MAPE | 13.0% | 17.4% | 15.8% | 16.7% | 14.0% | 13.2% | 10.5% | 10.3% | 10.3% | **9.9%** |

Table 5: (Correction of Table 2) Forecasting error (PMU).

|  | Metric | FNN | LSTM | DCRNN | LDS | NRI | GTSv | GTS |
|---|---|---|---|---|---|---|---|---|
| 15 min | MAE ($\times 10^{-3}$) | 1.23 | 1.02 | 0.71 | 0.49 | 0.66 | 0.35 | **0.26** |
|  | RMSE ($\times 10^{-2}$) | 1.28 | 1.63 | 1.42 | 1.26 | 0.27 | 0.22 | **0.20** |
|  | MAPE | 0.20% | 0.21% | 0.09% | 0.07% | 0.14% | 0.06% | **0.04%** |
| 30 min | MAE ($\times 10^{-3}$) | 1.42 | 1.11 | 1.08 | 0.81 | 0.71 | 0.45 | **0.37** |
|  | RMSE ($\times 10^{-2}$) | 1.81 | 2.06 | 1.91 | 1.79 | 0.30 | 0.29 | **0.26** |
|  | MAPE | 0.23% | 0.20% | 0.15% | 0.12% | 0.15% | 0.07% | **0.06%** |
| 60 min | MAE ($\times 10^{-3}$) | 1.88 | 1.79 | 1.78 | 1.45 | 0.83 | 0.63 | **0.59** |
|  | RMSE ($\times 10^{-2}$) | 2.58 | 2.75 | 2.65 | 2.54 | 0.46 | 0.43 | **0.41** |
|  | MAPE | 0.29% | 0.27% | 0.24% | 0.22% | 0.17% | 0.11% | **0.10%** |

Table 6: (Correction of Table 3) Forecasting error (PEMS-BAY).

|  | Metric | HA | ARIMA | VAR | SVR | FNN | LSTM | DCRNN | LDS | GTSv | GTS |
|---|---|---|---|---|---|---|---|---|---|---|---|
| 15 min | MAE | 2.88 | 1.62 | 1.74 | 1.85 | 2.20 | 2.05 | 1.38 | 1.33 | 1.35 | **1.32** |
|  | RMSE | 5.59 | 3.30 | 3.16 | 3.59 | 4.42 | 4.19 | 2.95 | 2.81 | 2.64 | **2.62** |
|  | MAPE | 6.8% | 3.5% | 3.6% | 3.8% | 5.2% | 4.8% | 2.9% | 2.8% | 2.9% | **2.8%** |
| 30 min | MAE | 2.88 | 2.33 | 2.32 | 2.48 | 2.30 | 2.20 | 1.74 | 1.67 | 1.69 | **1.64** |
|  | RMSE | 5.59 | 4.76 | 4.25 | 5.18 | 4.63 | 4.55 | 3.97 | 3.80 | 3.45 | **3.41** |
|  | MAPE | 6.8% | 5.4% | 5.0% | 5.5% | 5.4% | 5.2% | 3.9% | 3.8% | 3.9% | **3.6%** |
| 60 min | MAE | 2.88 | 3.38 | 2.93 | 3.28 | 2.46 | 2.37 | 2.07 | 1.99 | 1.99 | **1.91** |
|  | RMSE | 5.59 | 6.50 | 5.44 | 7.08 | 4.98 | 4.96 | 4.74 | 4.59 | 4.05 | **3.97** |
|  | MAPE | 6.8% | 8.3% | 6.5% | 8.0% | 5.9% | 5.7% | 4.9% | 4.8% | 4.7% | **4.4%** |

