# OpenReview forum: "Discrete Graph Structure Learning for Forecasting Multiple Time Series"
_ICLR.cc/2021/Conference — ICLR 2021 Poster_

### Official Review · AnonReviewer1 · 2020-10-26
**Simple, somewhat effective idea**

**Rating:** 6
**Confidence:** 3

**Review:**

The paper considers learning both graph structures and NNs for time series data, similar to the idea of LDS (Franceschi et al., 2019). Observing the computation and scalability issues with LDS, authors propose a unilevel optimization form wrt. the mean performance over the graph distribution. This is done via NNs, with input being the observed sequence, to output a real matrix whose elements are then treated as weights for the Gumbel trick. NN structures, training procedure, etc. mostly follow existing works.

Overall, the paper is well presented, easy to understand, with a simple and somewhat effective modification over LDS. I generally like simple ideas with sufficient insights and explanations (though there is not much in this work), but I'm not sure if the empirical improvement is sufficient. I recommend a weak acceptance for now and may change my score after reading other reviews.

I only have one question: the proposed idea is not restricted to time-series case. So how does it perform for non-time-series data? It would be a big benefit if the proposed idea also helps in a more general case than the present scope.

** after reading response ** I thank authors for replying to my question. I maintain the previous rating.

---

> ### Author Response · Authors · 2020-11-19
> **Response to question**
>
> Thank you for the encouraging review. To answer your question, we believe applying our method to cases outside time series incurs a new technical question. Time series data coincidently offers convenience for designing feature extractors needed in the structure learning (top part of Figure 1). In this case, there are ample tools to extract features for a
> long sequence. On the other hand, for data other than time series, the features of a node may need to be learned from a set/distribution of objects (e.g., documents, molecules, etc). Extracting features for a set may be less straightforward, depending on the actual use case.
>
> One also needs to adapt the bottom part of Figure 1 to a different GNN, if the node data is not a time series. But the adaptation should be relatively simple in light of the prosperous literature of GNNs.

---

### Official Review · AnonReviewer3 · 2020-10-28
**The proposed GTS appears to advance the current state-of-the-art in graph-based multiple (multivariate) time series forecasting. This is a problem of considerable importance and, as far as I am aware, simultaneously learning the graph structure and forecasting model is understudied topic. As for the several “Improvement points” I raised in this review, I believe that the authors will have the chance to address them in the rebuttal period.**

**Rating:** 7
**Confidence:** 4

**Review:**

Paper summary:

This paper proposes an approach for time series forecasting that learns the graph structure among multiple (multivariate) time series simultaneously with the parameters of a Graph Neural Network (GNN). The problem is formulated as learning a probabilistic graphical model by optimizing the expectation over the graph distribution, which is parameterized by a neural network and encapsulated in a single differentiable objective. Empirical evidence suggests that the proposed GTS obtains superior forecasting performance to both deep and non-deep learning based, as well as graph and non-graph based, competitor forecasting models. In addition, GTS appears to be more computationally efficient compared to LDS, a recently proposed meta-learning graph-based approach.

##########################################################################

Strong points:
1. A time series forecasting model is proposed to automatically learn a graph structure among multiple time series and forecast them simultaneously using a GNN.

2. The graph structure and the parameters of the GNN are learned simultaneously in a joint end-to-end framework.

3. The graph structure is parameterized by neural networks rather than being treated as a (hyper)parameter, thus significantly reducing the training cost compared with the recently proposed bilevel optimization approach LDS.

4. A structural prior-based regularization is incorporated in GTS. In case a “ground-truth” graph is provided upfront, this may serve as a healthy variation of such a graph for the purpose of more accurate forecast.

5. Extensive experiments are conducted in which the proposed GTS is compared to a number of baselines, including a recently proposed graph structure learning approach, and deep or non-deep learning based (as well as graph or non-graph based) forecasting models.

6. The experimental results demonstrate that GTS outperforms its competitor approaches in terms of forecasting accuracy and is more efficient that the recently proposed LDS.

7. Generally, the paper is well written, while the notation is clear and easy to follow.

##########################################################################

Improvement points:
1. In section 3.4. (Comparison with NRI), the authors state that the “structural prior” $A^{a}$ offers a stronger preference on the existence/absence of each edge than a uniform distribution over all edges. This seems a bit unclear, thus I would encourage the authors to elaborate a bit more on this difference between GTS and NRI w.r.t. the structural prior.

2. In the case of the PMU dataset, despite the fact that the grid topology is not provided, the authors still consider a certain structural prior by constructing a kNN graph among the PMUs. I am wondering whether the correlation between the series (mentioned briefly in Appendix A) is used for the graph construction or another distance/similarity metric is considered?

3. Two variables are recorded by the 42 PMUs, however each node in the constructed graph (shown in Fig. 5) corresponds to one PMU. In case a single node corresponds to a single PMU, then I wonder how the similarity between two PMUs’ recordings is calculated across the two variables (voltage magnitude and current magnitude)?

4. The authors construct the PMU dataset by extracting only one month of data. However, a single month of PMU data would not allow for capturing certain long-term seasonalities (for instance, the PMU recordings are typically impacted by outages that occur more frequently in certain seasons or periods in the year). Is this perhaps due to data unavailability? If that is not the case, I would ask the authors to clarify the reasoning behind the decision to extract the data for February 2017?

5. In Tables 2 & 3 (Appendix C), some of the MAPE values obtained by GST are bolded even though the same percentages are reported for GTSv. In such cases, I would suggest the authors to either bold the MAPEs obtained by both GTS and GTSv, or present the MAPE values using more decimal places.

6. There are several minor textual errors throughout the paper that can be easily addressed. Some of them are summarized as follows:
- The term “LDS” is initially used at the beginning of page 2, but is not defined earlier in the text.
- In the third paragraph on page 2, “computation is expensive” should be replaced by “its computation is expensive”.
- I am wondering whether the training loss $L$ should be used instead of the validation loss $F$ in Eq. (2)? If so, correct accordingly, otherwise disregard this comment.
- In the next-to-last paragraph on page 2, consider replacing “it is better scaled” with “it scales better”.
- In the last paragraph of the Related Work section, “of node classification tasks” should be replaced by “on node classification tasks”.
- At the beginning of section 3, the term “NRI” is used, but is not defined earlier in the text.
- On page 5, consider replacing the abbreviation “ELBO” with “evidence lower bound (ELBO)”.
- In the first paragraph on page 7, “treating it a (hyper)parameter as in LDS” could be replaced with “treating it *as* a (hyper)parameter *which is the case* in LDS”.
- In the second paragraph on page 8, replace “regularization $\lambda$” with “regularization strength $\lambda$”. In the same sentence, consider adding “the” before both “forecasting error” and “cross entropy”.

##########################################################################

Questions during rebuttal period:
Please address the aforementioned remarks/questions.

---

> ### Author Response · Authors · 2020-11-19
> **Revision and response, part 1**
>
> Thank you for the detailed comments. We have updated the paper accordingly. In what follows we respond point by point. Your comments are cited before the response.
>
> > In section 3.4. (Comparison with NRI), the authors state that the “structural prior” $A^a$ offers a stronger preference on the existence/absence of each edge than a uniform distribution over all edges. This seems a bit unclear, thus I would encourage the authors to elaborate a bit more on this difference between GTS and NRI w.r.t. the structural prior.
>
> NRI uses a variational autoencoder, wherein the latent prior is a uniform distribution in [0,1] for each node pair. Our structural prior is a given graph, each node pair of which either admits an edge (1) or not (0). We have edited the sentence.
>
> > In the case of the PMU dataset, despite the fact that the grid topology is not provided, the authors still consider a certain structural prior by constructing a kNN graph among the PMUs. I am wondering whether the correlation between the series (mentioned briefly in Appendix A) is used for the graph construction or another distance/similarity metric is considered?
>
> We used the $L_2$ distance metric. It is a somewhat arbitrary choice. We believe that correlation is equally meaningful; but even for domain experts, they do not have a consensus on the metric. The point here is to show that regularization is effective, no matter what structural prior one imposes.
>
> > Two variables are recorded by the 42 PMUs, however each node in the constructed graph (shown in Fig. 5) corresponds to one PMU. In case a single node corresponds to a single PMU, then I wonder how the similarity between two PMUs’ recordings is calculated across the two variables (voltage magnitude and current magnitude)?
>
> We simply concatenated the series for different variables to form a vector before computing the $L_2$ distance.
>
> > The authors construct the PMU dataset by extracting only one month of data. However, a single month of PMU data would not allow for capturing certain long-term seasonalities (for instance, the PMU recordings are typically impacted by outages that occur more frequently in certain seasons or periods in the year). Is this perhaps due to data unavailability? If that is not the case, I would ask the authors to clarify the reasoning behind the decision to extract the data for February 2017?
>
> We note two points. First, indeed we are constrained by data availability. The data set we have access to contains a huge amount of missing data (apart from other quality issues such as unreliability of recordings and inconsistent sampling frequencies). We selected a month that is relatively clean so that data quality does not adversely affect the evaluation of different methods.
>
> Second, a usual trick to accommodate seasonality is to subtract seasonal mean, so that the data looks more stationary. That said, we agree that if permitting, it is generally preferrable to include as many data for training as possible to accommodate long-term effects that cannot be satisfactorily resolved by simple preprocessing tricks.
>
> > In Tables 2 & 3 (Appendix C), some of the MAPE values obtained by GST are bolded even though the same percentages are reported for GTSv. In such cases, I would suggest the authors to either bold the MAPEs obtained by both GTS and GTSv, or present the MAPE values using more decimal places.
>
> Thank you for pointing out. We have bolded the equal values. We have also moved the tables to the main text because of extra space allowable for revision.

---

> > ### Comment · AnonReviewer3 · 2020-11-20
> > **Follow-up question**
> >
> > Thank you for addressing the raised comments. However, I do have a subsequent question regarding the following answer:
> >
> > > We simply concatenated the series for different variables to form a vector before computing the $L_2$ distance.
> >
> > I am wondering if concatenation is a reasonable choice since the range of the voltage magnitude values is different from that of current magnitude. Did you take this into account before concatenating the series for the two variables? For instance, did you perform a certain type of normalization so as to bring them on the same scale?

---

> > > ### Author Response · Authors · 2020-11-20
> > > **RE: Follow-up question**
> > >
> > > Indeed, as typical preprocessing, we standardize each variable (subtract mean and divide by standard deviation). Such normalization should bring the variables on the same scale.

---

> ### Author Response · Authors · 2020-11-19
> **Revision and response, part 2**
>
> Continued from part 1...
>
> > There are several minor textual errors throughout the paper that can be easily addressed. Some of them are summarized as follows: The term “LDS” is initially used at the beginning of page 2, but is not defined earlier in the text.
>
> Edited. The name “LDS” used in the original paper stands for a generic phrase “learning discrete structures”.
>
> > In the third paragraph on page 2, “computation is expensive” should be replaced by “its computation is expensive”.
>
> Edited.
>
> > I am wondering whether the training loss $L$ should be used instead of the validation loss $F$ in Eq. (2)? If so, correct accordingly, otherwise disregard this comment.
>
> The notations $F$ and $L$ are only function names; whether they mean training or validation loss depends on the third argument $X$.
>
> > In the next-to-last paragraph on page 2, consider replacing “it is better scaled” with “it scales better”.
>
> Edited.
>
> > In the last paragraph of the Related Work section, “of node classification tasks” should be replaced by “on node classification tasks”.
>
> Edited.
>
> > At the beginning of section 3, the term “NRI” is used, but is not defined earlier in the text.
>
> Edited. The name “NRI” refers to a generic phrase “neural relational inference”.
>
> > On page 5, consider replacing the abbreviation “ELBO” with “evidence lower bound (ELBO)”.
>
> Edited.
>
> > In the first paragraph on page 7, “treating it a (hyper)parameter as in LDS” could be replaced with “treating it as a (hyper)parameter which is the case in LDS”.
>
> Edited.
>
> > In the second paragraph on page 8, replace “regularization $\lambda$” with “regularization strength $\lambda$”. In the same sentence, consider adding “the” before both “forecasting error” and “cross entropy”.
>
> Edited.

---

### Official Review · AnonReviewer4 · 2020-10-28
**Official Blind Review**

**Rating:** 4
**Confidence:** 4

**Review:**

The paper proposes an approach for multivariate time series forecasting by trying to estimate dependence across dimensions via a learned graph structure.  The dimensions are considered as nodes in a graph, and the problem is mapped to learning a discrete graph structure that can help with downstream forecasting task.
The paper shows that a graph neural network (GNN) can be leveraged even though an explicit structure is unknown, to improve forecasting performance. This is achieved while learning the graph structure and forecasting architectures in an end-to-end fashion. The proposed approach is computationally efficient compared to a bilevel optimization approach where a discrete graph structure is learnt in a meta-learning framework. The approach is further claimed to be able to incorporate apriori knowledge of the graph structure by proposing a regularizer that ensures that the learned graph structure stays close to the known graph structure. The proposed approach improves forecasting performance in comparison to several strong baseline methods on three real-world datasets.
In general, the paper is well-written and easy to follow.

Attempts have been made in the past for learning such a discrete graph structure from data. The authors mention LDS [2] and NRI [3] as closest to their work. The authors attempt to explicitly compare the proposed approach to NRI. The authors claim that "The most essential distinction is the number of structures": one structure is learned in the proposed approach while many structures are learned in NRI. From what I could follow, this single structure is achieved by using the entire multivariate time series data to obtain a feature vector for each dimension (series) via a neural network instead of using window-wise data. In this sense, this appears to be a simplification of NRI, rather than being something novel and different.
The proposed setup and the approach are different and novel compared to NRI as the "Amortized inference is not desired nor relevant": I am not sure how this makes the proposed approach non-trivial given NRI?
Furthermore, in contrast to LDS, the key contribution of the proposed approach is to get rid of the bilevel optimization. But then, that also seems to rely mainly on the Gumbel reparameterization trick which has been used in NRI for forecasting albeit for a slightly different setting.

Another very closely related approach to the proposed one is that in [1], which the authors seem to be unaware of.
One of the main claims of the proposed approach is an attempt to learn the graph structure and forecasting model in an end-to-end learning fashion. However, this problem has already been attempted in [1].
Therefore, it is difficult to comment on the novelty and contribution of the paper without a comparison with [1], especially since most of the benchmark tasks and datasets used in this paper are present in [1] as well.

I have some concerns regarding the empirical evaluation:
1. Can the observations in Fig. 2 be attributed to the graph learning part? Despite the fact that the only difference between DCRNN and the proposed method seems to be the graph structure learning part, it is still not obvious qualitatively as to why the observations of Fig. 2 can be attributed to the graph learning part, e.g. why is "GTS curve better captures the sharp dip toward the end of the series" attributable to the graph learning part qualitatively or as per domain knowledge? I think an empirical analysis on a synthetic dataset to support such claims related to ablation could be useful.
2. In Fig. 4a, the regularization seems to induce sparsity and has been observed by the authors as well: "increasing the regularization magnitude produces sparser graphs". But the efficacy of such regularization on forecasting performance is not clear as k=0 (no kNN regularization) seems to have the best forecasting performance in Fig. 4a. This seems to imply that using kNN graph knowledge is not adding any value.
Similar observations can be made in Fig. 4b, where increasing $\lambda$ leads to increasing MAE. As such, the effect or usefulness of regularization is not clear.
3. The analysis on regularization and learned structures is done using a kNN graph as apriori knowledge for the PMU dataset. Rather than relying on another data-driven graph structure (kNN graph) as ground truth, I wonder if it would be useful to do such analysis on the public datasets (METR-LA and PEMS-BAY) for which the ground truth structures are actually known. As such, the evaluations on "effect of regularization" and "learned structures" do not seem conclusive.
4. In Tables 2 and 3, some of the bolds also apply to GTSv but are missing.

Given the above points, the originality of the work and the contributions are not clear.

Other minor points:
1. Related Work can also benefit from more precise references to papers that use the mentioned architectures for time series forecasting. The current references are too generic.
2. typo: hyperparemeter


References:
[1] Connecting the Dots: Multivariate Time Series Forecasting with Graph Neural Networks, Wu et. al, KDD2020. https://dl.acm.org/doi/abs/10.1145/3394486.3403118
[2] LDS: Learning discrete structures for graph neural networks, Franceschi et. al, ICML, 2019
[3] NRI: Neural relational inference for interacting systems. Kipf et. al, ICML, 2018.

---

> ### Author Response · Authors · 2020-11-19
> **Revision and debate**
>
> Thank you for the thoughtful comments. We have included in the paper a new set of experiments regarding the two traffic data sets (see the paragraph “Other structural priors” before Section 5). We would also like to take this opportunity to debate your assessment of similarity to prior work.
>
> NRI is related to our approach because the objective of inferring a latent graph from data is the same; both approaches use the graph to fit trajectories/series. However, NRI takes a variational Bayesian approach, where the graph is derived from posterior inference. Amortization is involved to reduce the inference cost. Our approach is not Bayesian at all. The concepts of “posterior”, “amortized inference”, and “prior” serve the purpose of bringing closer the interpretation of the two approaches, which otherwise are quite different in technical details.
>
> LDS is a bilevel optimization approach, whereas ours is unilevel. The key distinction does not lie in the use of the Gumbel trick. Rather, the distinction is whether to treat the graph as a free parameter or to parameterize it. Interestingly, this difference underlies the concept of amortized inference in variational Bayes, although we stress again that our approach is not Bayesian. Parameterizing the graph reduces the number of parameters and is more computationally friendly. However, this is not a free win: fewer parameters narrow the hypothesis space. Nevertheless, the tradeoff is rather intricate in practice, because treating the graph as a free parameter as in LDS imposes challenges on optimization.
>
> Encouragingly, we have empirically compared with NRI and LDS. The results all point to a better performance of our approach.
>
> Thank you for bringing to our awareness the very recent paper [1], MTGNN. Admittedly, we started our work before MTGNN became public. MTGNN shares similarity with our work as well as NRI in that all methods aim to infer a graph and use it to improve forecast. One difference between MTGNN and our work lies in the modeling of the graph structure. MTGNN parameterizes a degree-k graph, whereas our regularization accommodates any structure, admitting greater generality. (For example, the graphs in the traffic data sets we use do not have the same number of edges for each node.) We have updated the paper with this reference and a short discussion.
>
> Below are responses to your concerns on empirical evaluation.
>
> RE: Fig 2 (role of graph structure learning). Indeed, Fig 2 compares the use of a given graph versus a graph learned in our approach. Difference in performance directly comes from difference in the graph. The graph is a means for the neural network to aggregate information from other time series. The neighborhood defined by a given graph may not be optimal even if, for example, it is based on the spatial proximity. On the other hand, learning the graph offers opportunities to define a better neighborhood based on the supervised signal. What our approach learns proves to work better according to experimental results. Moreover, this finding is fairly consistent with that in LDS, which suggests that there is usually room to improve a given graph for node classification.
>
> RE: Fig 4 (role of regularization). Extending the rationale of the above argument, if there is room to improve the given graph, and because the regularization penalizes departure from it, then the regularization effectively produces something between the given graph ($\lambda=\infty$) and the optimal one (say, attained when $\lambda=0$), if optimality is measured by forecast quality. However, forecast quality may not be the sole aim one pursues. For example, one may tolerate a slight compromise in accuracy in exchange of a much sparser graph for better interpretability, or one may have a strong belief in a certain structure and desires only denoising. In such cases, regularization with a certain nonzero $\lambda$ is a means to encourage closeness to the a priori belief. Indeed, both Fig 4 and 5 convey a message that the actual choice of $\lambda$ depends on which end of the spectrum one desires.
>
> RE: Additional results on METR-LA and PEMS-BAY. We have updated the paper with experiments on these traffic data sets (see the paragraph “Other structural priors” before Section 5). The findings are very much similar to that of the PMU data set.
>
> Additionally, in the updated paper we have addressed miscellaneous issues you pointed out (boldface numbers, references, typos).

---

> > ### Comment · AnonReviewer4 · 2020-11-23
> > **Contribution of the work is not clear given prior art. A more thorough analysis would be useful.**
> >
> > Thanks for the response and the updated version of the paper.
> >
> > 1. Given NRI and MTGNN, I would still humbly argue that the novelty and contribution of this work is not clear, as detailed in the original review.
> > Apart from the concerns regarding the differences of NRI and the proposed GTS, the authors claim that the proposed GTS is empirically better than NRI: "Encouragingly, we have empirically compared with NRI and LDS. The results all point to a better performance of our approach." The comparison with NRI is based on results on only one dataset, where NRI, GTSv and GTS all have extremely low errors (e.g. MAE of the order of 10^-4, and MAPE of less than 0.2%.) The results of NRI on other datasets are not provided due to memory consumption issues.
> > Perhaps, a more thorough analysis on a few synthetic datasets with varying characteristics could have been useful if memory consumption issues do not allow evaluation on the other two datasets.
> > Similarly, an empirical comparison with MTGNN would allow to evaluate the claim on the usefulness of generality of GTS over MTGNN "we on the other hand allow a more general structural prior for the graph".
> >
> > 2. I think the concern regarding Fig. 2 still remains unaddressed: why is "GTS curve better captures the sharp dip toward the end of the series" attributable to the graph learning part, either qualitatively or as per domain knowledge? I think an empirical analysis on a synthetic dataset to support such claims related to ablation could have been useful.
> >
> > 3. Thanks for adding the analysis is Fig. 6. It does shed light on the effect of regularization. However, given that regularization always hurts forecasting performance, and that the paper does not thoroughly evaluate structure learning as the focus of the paper is not to unearth the ground-truth structure, it's role is still not clear in the overall context.
> > The paper mainly evaluates forecasting in comparison to other methods, and clearly states that unearthing the ground-truth structure is not the focus at multiple places. But then, the authors suggest that "both Fig 4 and 5 convey a message that the actual choice of lambda depends on which end of the spectrum one desires." and "However, forecast quality may not be the sole aim one pursues."
> > This complicates the scope of the work, and accordingly, the evaluation.
> > A useful direction in this context could be to explore if the prior knowledge of graph helps with faster/robust training in case of limited training data.
> >
> > A typo: forest --> forecast

---

> > > ### Author Response · Authors · 2020-11-24
> > > **Debate on contribution**
> > >
> > > Dear reviewer, your comments are well taken but you may have missed several points.
> > >
> > > Differences from prior work lead positively to the improved empirical results. Such an improvement underlies the contribution of the work, as it points to updated knowledge of what design probably works better than others.
> > >
> > > The fact that experiments with NRI are limited to one data set reveals the scalability challenge of NRI, because the structure learning therein uses message passing on a complete graph. This fundamental limitation cannot be resolved as you suggest by doing more experiments on synthetic data sets.
> > >
> > > The absolute errors on PMU are small because the original values are small. It makes little sense to read the magnitudes; the more important message comes from the error drop from method to method. For example, the top row of Table 3 reads: 1.23, 1.02, 0.71, 0.49, 0.66, 0.26/0.24. The last two numbers 0.26/0.24 are for GTSv/GTS; they are noticeably much better than previous values.
> > >
> > > The point of this paper is not to say that we do not care about the learned structure and solely chase the best forecast. On the contrary, it was the observation that the learned structure without regularization was so uninterpretably dense that motivated us to formulate regularization in the first place.
> > >
> > > The regularization is rather effective. See, for example, the newly added Figure 6. On METR-LA, taking $\lambda$ to be 0.3 or 0.5 compromises barely the MAE, yet the cross entropy drops under 1.1e-1. Such a cross entropy indicates that the learned probabilities in $\theta$ are on average 0.1 away from the entries of the provided $A^a$, which is binary. The specific trade-offs between MAE and cross entropy vary across data sets, but we believe the message that “the learned structure can be very close to the a priori belief with at most a slight degradation in forecast quality” unarguably holds. This message is in conjunction with the finding that even with a small compromise, the accuracy is still substantially better than all compared methods. We believe such empirical findings are strong.
> > >
> > > With the above findings, the advantage of our method over MTGNN is clear: we can get structures we desire, not restricted to a degree-k graph. This advantage is an important contribution, because MTGNN hard-wires the graph parameterization and enforces that all nodes have the same number of (out) neighbors. What if one desires instead a graph that approximately obeys spatial proximity, like that in the case of METR-LA? Our method yields such a graph.
> > >
> > > When interpreting Fig. 2, we say “the GTS curve better captures the sharp dip toward the end of the series.” This is an observation, not a claim. The intention here is to describe the results. We don’t know if, and we don’t feel, GTS will better capture *every sharp dip*. The overall lower error already indicates that our prediction curve is closer to the ground truth. The only methodological difference that causes this observation is that we change the supplied graph to the learned graph. A finer-grade ablation does not seem to be sensible. Furthermore, synthetic data serves no purpose here.
> > >
> > > Overall, we stress that this work contributes a method to learn a graph structure that helps better forecast. The forecast quality is improved over the absence of a graph and over the use of an a priori graph. In the latter case, the learned structure can be quite close to the a priori graph, while maintaining noticeably better forecast quality. The use of such a graph for regularization renders sparsity or other desired structures that admit good interpretability.

---

### Decision · Program_Chairs · 2021-01-07
**Final Decision**

**Decision:**

Accept (Poster)

**Comment:**

This paper presents a graph neural network-based approach to forecasting multiple time series. It incorporates structure learning similar in some ways to NRI (Kipf et al.) and a recurrent graph convolution forecaster given the inferred graph based on DCRNN (Li et al.). The paper shows consistently improved performance over several different kinds of baselines (classical, deep learning, and graph-based deep learning) on three datasets, two of which are public and one proprietary.

Reviewer 1 thought the paper was “well presented and easy to understand”. I agree. The reviewer liked the simplicity of the approach but wondered whether the empirical improvement was sufficient. The reviewer asked about applicability outside the time series domain and the authors provided a satisfying response.

Reviewer 3 thought that simultaneously learning graph structure and forecasting was an “understudied topic”. The reviewer pointed out several strengths including: the end-to-end nature of the approach, the reduction in training cost due to the direct parameterization of the adjacency matrix structure, the optimal structural regularization scheme, and the extensive experimentation. Like R1, they thought the paper was well written. They suggested several points of improvement, which were mainly seeking clarification. They made a good point regarding the PMU dataset in that only one month was considered, which was not enough to capture long-term seasonalities. This seems like a limitation to me. The authors responded to each of the points in turn. Regarding the point about the PMU dataset, the authors stated that the data was extremely noisy, so they settled on a month that was comparatively clean.

The review from R4 was not as positive as the other reviews. R4 proposed that the present work may be a “simplification of NRI” rather than being novel/different because NRI was a windowed-based approach, while GTS was based on the entire time series. The reviewer also pointed out what appears to be a highly relevant paper (Wu et al.); though this appeared in KDD 2020 and in my opinion it’s understandable if the authors missed it. Finally, R4 raised several issues with the empirical evaluation. They make a very good point that the analysis on regularization used the kNN graph where a “ground truth” graph was not available. Why not evaluate the regularizer on the other datasets where ground truth is available? The authors responded with an updated paper addressing this point. The authors responded to other points of criticism and a fairly extensive debate ensued.
The key points of the debate were:
A difference in opinion between the significance in departure between the structure learning mechanism in this work (GTS) and NRI (Kipf et al.)
Overlap between a recently proposed paper (Wu et al., KDD 2020). I am fairly sympathetic with the authors here. That work is fairly late-breaking and they do point out a key difference: "the advantage of our method over MTGNN is clear: we can get structures we desire, not restricted to a degree-k graph. This advantage is an important contribution, because MTGNN hard-wires the graph parameterization and enforces that all nodes have the same number of (out) neighbors. What if one desires instead a graph that approximately obeys spatial proximity, like that in the case of METR-LA? Our method yields such a graph."
Some minor concerns with Fig. 2 and the structural prior/regularization analysis

I have read the paper, and while R4’s concerns are legitimate, I think this paper is clearly over the bar. I support this paper’s acceptance and ask the authors to take the reviews into consideration when revising their paper. As R4 suggests, they could add a controlled set of experiments on a perhaps synthetic dataset to show that the proposed GTS is better than NRI. If the scalability of NRI is a concern, then this can be highlighted to the same extent as LDS.